# Three-phase electric power driven electoluminescent devices

Junpeng Ji[1], Igor F. Perepichka [2], Junwu Bai[1], Dan Hu[1], Xiuru Xu[1,3], Ming Liu[1], Tao Wang[1], Changbin Zhao[1], Hong Meng [1✉] & Wei Huang [2✉]

Current power supply networks across the world are mostly based on three-phase electrical systems as an efficient and economical way for generation, transmission and distribution of electricity. Now, many electrically driven devices are relying on direct current or single-phase alternating current power supply that complicates utilization of three-phase power supply by requiring additional elements and costly switching mechanisms in the circuits. For example, light-emitting devices, which are now widely used for displays, solid-state lighting etc. typically operate with direct current power sources, although single-phase alternating current driven light-emitting devices have also gained significant attention in the recent years. Yet, light-emitting devices directly driven by a three-phase electric power has never been reported before. Benefiting from our precious work on coplanar electrodes structured light-emitting devices, in this article we demonstrate proof of a concept that light-emitting components can be driven by three-phase electric power without utilizing intricate back-end circuits and can compose state detection sensors and pixel units in a single device inspiring from three primary colors. Here we report a three-phase electric power driven electroluminescent devices fabricated featuring of flexibility and multi-functions. The design consists of three coplanar electrodes with dielectric layer(s) and light emission layer(s) coated on a top of input electrodes. It does not require transparent electrodes for electrical input and the light emission occurs when the top light-emitting layers are connected through a polar bridge. We demonstrate some applications of our three-phase electric power driven electroluminescent devices to realize pixel units, interactive rewritable displays and optical-output sensors. Furthermore, we also demonstrate the applicability of three-phase electrical power source to drive organic light-emitting devices with red, green and blue-emitting pixels and have shown high luminance (up to 6601 cd/m$^2$) and current efficiency (up to 16.2 cd/A) from fabricated three-phase organic light-emitting devices. This novel geometry and driving method for electroluminescent devices is scalable and can be utilized even in a wider range of other types of light-emitting devices and special units.

[1] School of Advanced Materials, Peking University Shenzhen Graduate School, 2199 Lishui Road, Shenzhen 518055, China. [2] Institute of Flexible Electronics, Northwestern Polytechnical University, 127 West Youyi Road, Xi'an 710072, China. [3] College of Mechatronics and Control Engineering, Shenzhen University, 3688 Nanhai Street, Shenzhen 518000, China. ✉email: menghong@pku.edu.cn; iamwhuang@nwpu.edu.cn

Smartness, energy conservation, and multi-functions are three essential requirements for illumination systems and display devices. Light-emitting elements accompanied with soft electronics can be used for various applications, such as panel displays[1–3], artificial skins and muscles[4–6], optical sensors[7–10], and wearable electronics[11–14]. To meet the requirements for these applications, many types of light-emitting devices have been extensively studied, such as organic light-emitting devices (OLEDs), polymer light-emitting devices, perovskite light-emitting devices, quantum dot light-emitting devices, and inorganic light-emitting devices (LEDs). The researchers mainly focus on the light-emitting materials and structures of such devices, whereas there are very few studies on their driving sources. Recently, alternating current (AC)-driven electroluminescent (EL) devices have attracted increased attention and are regarded as promising alternatives to traditional direct current (DC)-driven EL devices[15,16]. There are several fundamental reasons for that. First, for injection DC-driven EL devices operated at high current density, the electroluminescence is markedly limited by triplet–triplet or triplet–charge annihilation. In contrast to that, the continual reversal of applied electric field in AC-driven EL devices can help to avoid charge accumulation, which may reduce triplet-exciton annihilation at high current densities[16,17]. Second, the introduction of an insulating dielectric layer can prevent electrochemical reaction between the electrode and the emissive layer, protecting the device degradation from the moisture and oxygen in the atmosphere[18–20]. Finally, and even more importantly, AC/DC converters and other costly switching devices must be built up for DC-driven EL devices that introduces power losses and additional complicated back-end electronics. In contrast, AC-driven EL devices can be easily integrated into 110/220 V and 50/60 Hz AC power lines without these drawbacks. It should be mentioned, however, that traditional AC power-driven LED systems always need very intricate design[21–23].

In our previous work[7], we have demonstrated a single-phase AC EL device with coplanar electrodes and realized long-distance optical communication. However, the two-electrode system limited some further applications. If we can use three or more electrodes with corresponding light-emitting layers above them, we can expand the colors of emission and realize the function of pixels or full-spectrum light emission. Thus, three-phase (TP) electric power, instead of single-phase AC power is used in almost every country across the world for AC power generation, transmission, and distribution, because TP system have advantages in all these aspects (Supplementary Note 1)[24,25]. Although almost all power supply networks currently use TP systems, some electrical equipment is not suitable for TP power and is only suitable for single-phase power. As the motor components[26] and the electric heating components[27] can be easily designed to be integrated into the TP electric system, light-emitting components become the most important components that cannot yet be adapted to the TP AC electric system. The existing solution is to distribute the light-emitting elements as evenly as possible into different single-phase lines. However, such light-emitting components, especially for large landscape lighting connected to the power grid can cause TP current imbalance. Unbalanced currents produce different voltage drops in each phase of the system, resulting in a TP imbalance of the system voltage and causing serious problems in power quality (PQ). Annual cost due to poor PQ caused by factors such as TP imbalance of voltage represents over €25 billion in the European Union economy[28]. Using TP driving electrical equipment should be a viable solution to avoid these losses. TP light-emitting sensors can be integrated into long-distance transmission lines to show the running condition and remotely warning optically. Furthermore, the phase difference in a TP power can provide an extra variable to drive light-emitting devices and will provide some new methods to control pixels. However, to the best of our knowledge, there are no currently reported light-emitting devices driven directly by TP electric power without using any switching mechanisms. Therefore, it seems to be important to fill in this gap and investigate the possibilities of using TP power for driving light-emitting devices.

In this study, we combined the advantages of TP electric power and AC EL devices, and demonstrated a new structure of TP electric power-driven EL (TPEL) devices. We demonstrated a series of light-emitting components using this new structure for different applications such as pixel-formed TPELs and multi-functional TPEL panels. The TP driving system and externally coupled polar bridge can provide rich applications. Furthermore, we extended the TP driving method to OLED systems to achieve high luminance and efficiency and confirmed that the TP driving method is widely applicable to various luminescent materials. The proposed novel concept of design of EL devices shows great potential for illumination and pixel formation.

## Results

**Fabrication and characterization of TPEL devices.** The fabrication process for a standard TPEL device is shown in Fig. 1a. A piece of conductive indium tin oxide (ITO) film using glass or polyethylene terephthalate (PET) as a substrate was uniformly divided into three parts as three coplanar electrodes by a laser etching process. The dielectric layer, consisting of 1 : 1 (by weight) mix of $BaTiO_3$ nano powders ($d < 1\,\mu m$) and commercial binder (EL binder 026, Nanjing Collaborative Innovation Lighting) was deposited on the etched ITO films using a blade coating method. The dried dielectric layer was sequentially blade coated with a light-emitting layer, which was constituted by blending commercially available ZnS : Cu phosphor powders (either GG45 (phosphor 1, green), GG14 (phosphor 2, orange) or GG65 (phosphor 3, blue), Leuchtstoffwerk Breitungen GmbH), and commercial cyanoresin binder (EL binder 028, Nanjing Collaborative Innovation Lighting) in 1 : 1 ratio (by weight) (Fig. 1d). According to our previous studies[7], polar electrode bridge (PEB) on the top of the phosphor layers, consisting of transparent polar liquid (such as water, ethanol, or other polar solvents) or polar solid (e.g., gel electrolyte), bridging between all the electrodes is needed to obtain light emission. Unless otherwise stated, we used deionized (DI) water as the polar bridge for device operation, as it is optically transparent, totally non-toxic, and easily available. The entire fabrication process was exposed to air, with no harsh preparation conditions were required and can be easily used for large-area fabrication. The schematic diagram and photographic image of working TPEL device are as shown in Fig. 1b, c. According to the cross-sectional scanning electron microscopy image (SEM) and corresponding energy dispersive X-ray spectroscopy map (Fig. 1d and Supplementary Fig. 1), the thicknesses of the dielectric layer using a blade coating method were about 15 µm as an average, which is somewhat thinner than using a screen-printing method reported in ref. [7] (~20 µm). A thinner dielectric layer is beneficial to increase the electric field intensity between the phosphor layer and the electrodes according to Supplementary Note 2 and Supplementary Fig. 2.

Traditional EL structures can be divided into two main categories: top-emission structure (TES) and bottom-emission structure (BES) (Supplementary Fig. 3). The essential difference between TES and BES structures is whether the light emitted from the light-emitting layer passes (or not) through the substrate[29]. Although TES devices have smaller light loss due to shorter luminous path and do not require transparent substrate, the light emitted from such devices also need to go through the transparent top electrode. The introduction of transparent

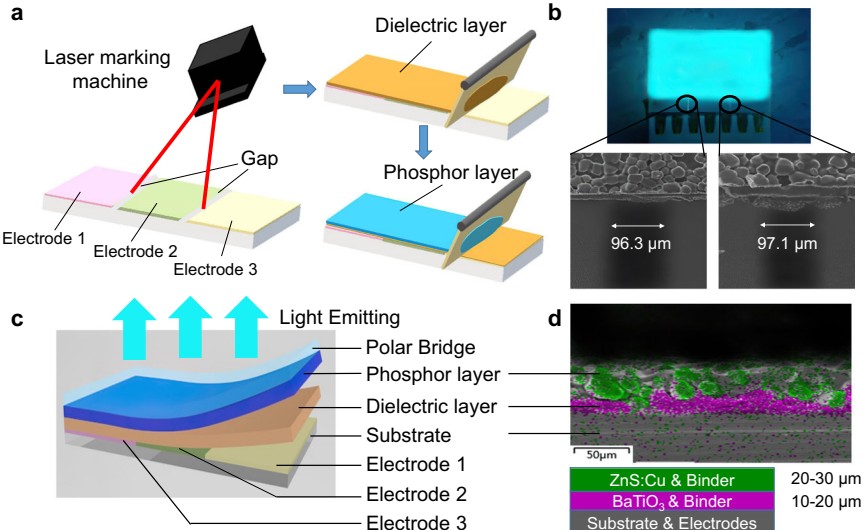

**Fig. 1 Schematic illustration of the preparation process and characterization of TPEL devices. a** Fabrication process of a standard TPEL device, including three electrodes preparation with gaps formation between them by laser marking and blade coating the dielectric and phosphor layers. **b** The photograph of working TPEL device (3 × 2 cm) operated with DI water polar bridge. Bottom figures show SEM images of the inter-electrode gaps. **c** Schematic "exploded view" of a TPEL device with three separated coplanar electrodes, a dielectric layer and a light-emitting layer blade coated above the electrodes. A polar bridge is not an inherent part of the TPEL device but necessary for light emission. **d** Cross-sectional energy dispersive X-ray spectroscopy (EDS) map of a TPEL device, with its schematic representation on the bottom. The colors on the EDS image correspond to that on the bottom scheme. The green and purple parts on the map represent the elements Ba and Zn, respectively, whereas the gray color in a phosphor layer and dielectric layer is a binder.

electrodes not only greatly complicates the fabrication process but also doubled the cost. It should be noted that the structure of TPEL devices we used in this work is an expanded version of polar-electrode-bridged EL light source (PEB-EL) as reported in our recent work[7]. We full exerted the advantages of such a structure and successfully fabricated new EL devices driven by TP electric power, while remaining the characteristics of the PEB-EL, which needs neither transparent electrodes nor tailored substrate. To prove the universality of our TPEL devices, we demonstrated TPEL devices fabricated using no substrate and ubiquitous electrodes such as tin foils and copper wires (Supplementary Fig. 4 and Supplementary Movie 1).

We measured the TP voltage and corresponding current waveforms of the device (Fig. 2a), and observed an obvious capacitive property in the TPEL devices with the currents to be ahead of the voltages by ca. 35° (Supplementary Note 3). Figure 2b demonstrates the relationship between the light output and the period of driving voltage for each phase, together with the total luminance from the TPEL device. As seen, the light emission from a single phase has a certain delay compared to the excitation by applied voltage and the luminance reaches its maximum when the voltage of each phase is at its negative period. The total emission from the TPEL device becomes smoother over the time as compared to the emission from a single phase.

This observation can be explained as follows. A TPEL device represents three coplanar capacitors, which are turned into parallel plate out-of-phase capacitors when the polar bridge electrode is attached on the top of the phosphor layer (Fig. 1b and Supplementary Fig. 2a, b). This forms a set of parallel capacitors connected in a series, with the phosphor layers (Ph) to be sandwiched between the electrodes (E1–E3) and the PEB (i.e., E1/Ph/PEB…PEB/Ph/E2, E2/Ph/PEB…PEB/Ph/E3, E3/Ph/PEB…PEB/Ph/E1). In this case, PEB can be regarded as an effective top electrode above the phosphor layer connecting to three power supply electrodes shifted by a phase of 120°. During an AC excitation, the direction of the dipoles in PEB above the phosphor is constantly reversed, inducing an electric field perpendicularly

to the phosphor layer. With some reservations, such configuration can be considered to be related to double-insulation AC-driven tandem LEDs[15]. We should also mention recent paper on liquid-interactive ferroelectric sound devices based on the planar AC architecture with a top liquid electrode, in which the vertically oscillating AC field generates sound from a piezoelectric layer and utilized for the detection of the polarity of the top liquid polar bridge[30].

The general mechanism of light emission in inorganic TPEL devices with PEB bridge is related to the scenario observed in a capacitive single-phase driven PEB-EL light sources reported by us previously[7], with the difference that we have now several capacitors, phase-shifted by 120°. In the first half-cycle of AC excitation in one of three capacitors with a polarization E(+)/Ph/PEB(−) causing by a phase shift in TPEL, charge carriers are generated in the charge generation/recombination zone of the semiconductor, and electrons and holes in the phosphor layer travel in opposite directions toward PEB(−) and E(+), respectively. Holes (being less mobile) are captured by the $Cu^+$ luminescence centers, whereas electrons (being more mobile) travel and are caught in shallow traps with no light emitted in this half-cycle (Fig. 2b). When the voltage is reversed in next half-cycle to E(−)/Ph/PEB(+), electrons generated in the first half-cycle are accelerated to the other side and radiatively recombine with holes at $Cu^+$ sites of ZnS : Cu luminophore causing EL emission.

The two other E(+/−)/Ph/PEB(−/+) capacitors also work in the same manner of cycling with a phase shift of 120°. This means that when the polarization of the first capacitor is, for example, E1 (+)/Ph/PEB(−), the second capacitor has an opposite polarization of E2(−)/Ph/PEB(+) with an opposite direction of charge travels across the phosphor (e.g., when E1 = + 150 V, the applied voltage E2 = −75 V, Fig. 2b) and vice versa. Also, as no charges are injected from the electrodes (because they have been separated from the emitting layer by dielectrics), the device shows electrode-independent characteristics, in contrast to charge-injection devices (e.g., DC-driven OLEDs). It should be noted that in the steady-state conditions under the continuously

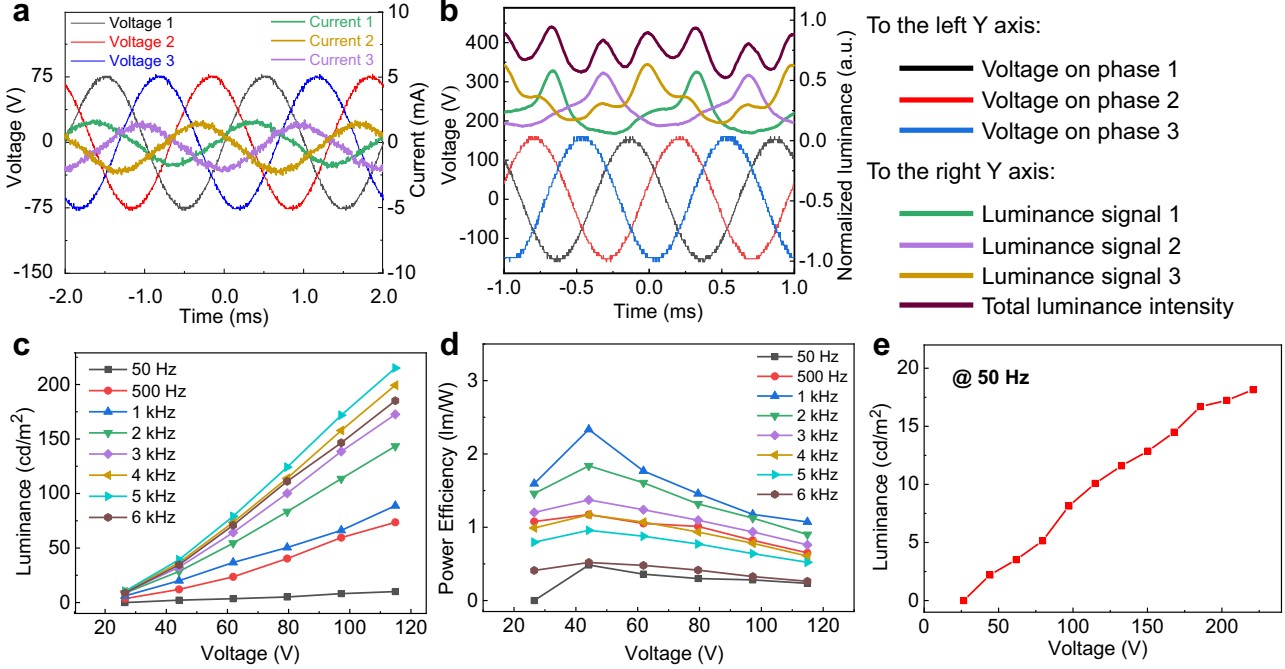

**Fig. 2 Characterization of TPEL devices. a** Oscilloscope signals of the three-phase voltage at $V_{rms} = 53$ V, 500 Hz and corresponding three-phase current oscillograms. **b** Relationship between the periods of three-phase driving voltage ($V_{rms} = 106$ V, 1000 Hz) and the light output for each phase and the total light output intensity for standard TPEL device using DI water as a polar bridge. Luminance (**c**) and power efficiency (**d**) of the TPEL device at different applied frequencies as a function of applied voltage. **e** Dependence of luminescence on applied AC voltage of 50 Hz frequency, showing that our TPEL devices can be driven directly by mainstream power supply system of 110/220 V, 50/60 Hz. All the above measurements were conducted using a TPEL device (3 × 2 cm) with glass-ITO as a substrate/electrodes, commercial phosphor GG65 as an emitter, and DI water as a polar bridge. Source data are provided as a Source Data file.

applied TP AC voltage, there is three times more voltage reverse in a TPEL device than a single-phase electric power-driven device in one period with the same frequency, and the charge recombination occurs in both half-cycles for the whole TPEL device. Moreover, at the period of time when one of the capacitors is not emissive, two other capacitors emit the light. Therefore, the total emission time of a TPEL device is longer than that of conventional single-phase AC EL device, thus leading to a rather continuous electroluminescence of the whole TPEL device in a time (Fig. 2b, dark brown line for the total luminance from a TPEL device).

Although several theories have been proposed in the literature to explain functioning the ZnS-powder AC EL devices, the mechanism of light emission is not is not fully understood and widely agreed upon, still provoking many debates[31–36]. In a doped ZnS : Cu semiconductor, because of low solubility of $Cu_xS$, it precipitates from the ZnS host during synthetic processing. Being p-type and n-type semiconductors, respectively, they form *p–n* heterojunction and a formation of these nano-precipitates is crucial for charge generation, separation, and recombination with an emission from the phosphor.

One of the most popular model of bipolar field emission by Fischer[31] has been that $Cu_xS$ forms the needles inside the ZnS particles (Supplementary Fig. 5). When an electric field is applied, relatively high fields will be concentrated on the tips of $Cu_xS$ conducting needles to induce tunneling the charges to the opposite sides of the needles to ZnS lattice and their trapping, following charge recombination in the reverse cycle to produce EL (Supplementary Fig. 6). In details, this model contradicts with some other observations reported in the later years, e.g., exclusively near the surface EL[35] and an importance of orientation of the particles to the applied field[36]. Thus, it was demonstrated that an emission is not uniform through the whole

ZnS : Cu particles (20–30 μm), but appears as small (1–2 μm) bright dots near the surface and associated with $Cu_xS$ precipitated from ZnS host[34,35]. It was concluded that while the emission occurs in both half-cycles of polarization, the difference in the luminance is due to non-transparency of ZnS and non-reflective character of the insulating layer, so the spots are brighter when the emission occurs from the side of an observer[35]. Yet, independently on what are the intimate details of the mechanism in single-phase AC-driven EL devices with ZnS : Cu phosphor, this has no effect on consideration of the features of an operation of capacitive TPEL devices.

We compared the luminance performance of the same TPEL device driven by a TP, single-phase, and sandwich electric power (the driving methods are shown in Supplementary Fig. 7a–c). Supplementary Fig. 7d compares the luminance performance of the device driven by the three methods as a function of a phase voltage, demonstrating a great improvement in luminance for the device operating under the TP driving system. It should be noted that actual voltage between the three electrodes is a line voltage for TP driving system. Therefore, we converted the voltage X axis into the line voltage by multiplying the phase voltage by √3 (for a single-phase system, it remains unchanged). However, even with the corrected voltage axis, an obvious improvement in the device performance was observed for the TP driving system as compared to the single-phase and conventional sandwich power systems (compare the luminance for the devices, Supplementary Fig. 7e). This can particularly be attributed to the rather continuous luminescence of the whole TPEL device (as seen from Fig. 2b), the luminance of which is smoother over the time compared to the devices driven by a single phase in PEB or sandwich configurations (Supplementary Fig. 7b, c).

Figure 2c, d and Supplementary Fig. 8 show the EL performance and the efficiency of our TPEL devices as functions

of applied voltage at different frequencies. In our previous work, we showed that various polar organic and aqueous liquids can be used as a polar bridge and compared the dependence of luminescence intensity for different polar bridges[7]. Thus, we have chosen the well-performing DI water as PEB in our experiments. Similar to single-phase-driven PEB-ELS devices, the luminescence efficiency increased with the applied voltage reaching a maximum value of 215 cd/m$^2$ at 5 kHz and 115 V. On further increasing the frequency, the luminescence decreased. Although higher frequency gives more probability for charge generation, separation, and recombination at the copper luminophore centers for light emission within a certain time, when the AC alternation is too fast, the process of electrons/holes generation and separation becomes less efficient, resulting in decreased luminescence efficiency. Supplementary Fig. 9 shows the effects of frequency on the performance of the TPEL device. The power and current efficiency were increased with the AC frequency reaching a maximum at about 1 kHz and then dropped down. From this, we can infer that the lifetime of the excited luminophore should be around 1 ms. When the cycle time is longer than the lifetime of the excited luminophore (frequency < 1 kHz), increasing the frequency provides more chances to produce an excited state of the luminophore. However, when the cycle time become shorter than the lifetime of the excited state of the luminophore (frequency > 1 kHz), lesser excitations make the efficiency lower[7].

It is worth noting that our TPEL devices can be driven directly by the mainstream TP power supply systems such as 110/220 V, 50/60 Hz (Fig. 2e). We also observed an obvious and continuous color changing phenomenon with the change of frequency (see also ref. [2]). Thus, when the frequency was increased from 50 Hz to 1000 Hz, the emission color of GG65 was gradually changed from green to blue region (from CIE (x,y) = (0.280, 0.467) to (0.167, 0.275), Supplementary Fig. 10).

**A novel concept for next-generation pixel formation.** "The Tao produced One; One produced Two; Two produced Three; Three produced All things"[37]. This famous quote, told by the Chinese ancient philosopher Tzu Lao seems to be suitable to describe the full colors display field, because using three primary colors we can create any color on the palette. As the TPEL device has three light-emitting planes, it can be easily designed as pixels, which emit three different colors. Traditional pixel must be decomposed into three devices: red light-emitting device, green light-emitting device, and blue light-emitting device. Every device need two electrodes and one of them must be transparent. Thus, expensive transparent electrodes are essential parts for traditional pixel formation. Here we demonstrate a potential novel concept for the next-generation pixel formation by using a designed TPEL device. Using this concept, we can save half of the electrodes and no expensive transparent electrode is needed to form a pixel model. The schematic diagram of such three-color TPEL device is shown in Fig. 3a. We used different commercial phosphor powders to realize light emission of different colors (Fig. 3b). In contrast to standard TPEL devices described above, in a pixel-formed TPEL, we introduced a ground electrode without coating any phosphor layer and used TP four-wire system for controlling the switch of each electrode independently. A pixel-formed TPEL device was easily controlled by three switches and four wires to simulate all luminous state of pixels (Fig. 3c and Supplementary Movie 2).

Such pixel-formed TPEL devices can also be separated to be relatively independent (Fig. 4a–c and Supplementary Movie 3). In this configuration, no ground electrode was used. Three devices with different phosphors connected to three AC power lines were placed in separate beakers with DI water as PEB. When DI water

PEBs were connected together via hydrogel to form a bridge, each device emitted the light from its phosphor. Besides relatively poor conductor such as hydrogel, conductive metal can also be a part of a polar bridge (Supplementary Fig. 11 and Supplementary Movie 4), proving that high electrical conductivity is not a decisive factor for device operation.

**Luminous panel for interactive writable display and large-area landscape lighting.** In contrast to point light sources, luminous panels such as AC EL panels and OLED panels are featured by soft uniform light with no eye injury, power saving, naturally occurring lighting, etc. When the surface light sources are applied for general lighting or public displays, we usually need them to be capable to apply for large emitting area. This might cause the TP imbalance if we drive such large-area light-emitting devices by a single-phase AC electric power when connected to the power grid. Here we demonstrate a concept for large-area luminous panels driven by TP electric power, in which case the phase imbalance of the power grid is avoided. We have designed two different types of electrode patterns applying them to large-area TPEL devices, which we call lollipop-type and triangle-type electrodes, respectively (Fig. 5a, b). Such patterns are alternatives to traditional interdigital electrodes when it comes to TP driving method. Such electrode patterns, however, extend beyond our TPEL devices and can be applied to all situations where we want to use interdigital electrodes in a TP system.

As an example, the schematic diagram of TPEL panels using lollipop-type electrodes is shown in Supplementary Fig. 12. Their fabrication process is similar to that for standard TPEL devices and the TPEL panels working with DI water as a polar bridge are demonstrated in Fig. 5c, d. It is worth noting that our TPEL panels are applicable not only for general solid-state lighting, but can also be used in other technologies, e.g., as interactive rewritable displays. An erasable interactive rewritable display has been fabricated and writing symbols on its top surface by water using wet brush (or by commercial water-based fluorescent pen), their erasing and re-writing have been successfully demonstrated (Fig. 5e and Supplementary Movie 5). In this case, water from the brush (or other kind of writing ink) acts as a polar bridge, so only the part of the panel covered by a polar bridge starts to emit the light. Unlike other interactive rewritable displays, our TPEL panel not only driven by TP power, but also needs neither special conductive and transparent material[1] nor complex pressure-sensing system and back-end circuit[11,38]. Such TPEL-based interactive panel can be easily "initialized" by drying the polar bridge (e.g., by absorbent paper) with new writing on this rewritable display.

Many more applications are seen for such TPEL panels. To demonstrate their advantages of direct connection to 110/220 V, 50/60 Hz power lines, we showed applications in the state detection of long-distance electric transmission lines and optical alarm remotely. Almost all the factors that threaten the safety of power lines such as rain, snow, dew, frozen rain, ice accretion, or extremely wet environment can be detected and raise an optical alarm remotely by the connected TPEL panels (Supplementary Fig. 13). The TPEL panel with implemented polar bridge can also work as a sensor remotely communicating optically when exceptions occurring in the power lines, such as phase loss or TPs imbalance (Fig. 5f). Bendable TPEL panels can also be fabricated using flexible substrates such as PET films (Fig. 5g), thus expanding their potential applications.

**Fabrication and characterization of TP-OLEDs.** AC-driven OLEDs have a long history of over two decades[39,40] and different types of the device structures have been proposed and studied[15,

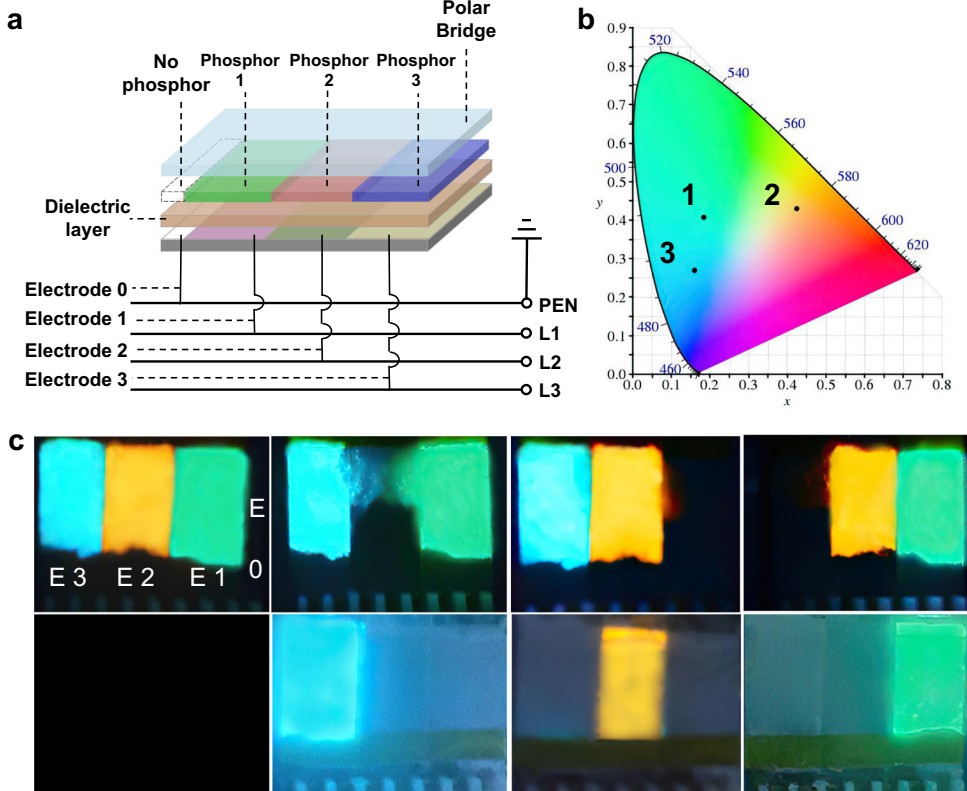

**Fig. 3 Schematic illustration of the structure and the control method of the pixel form TPEL devices. a** Schematic diagram and connection method using a three-phase four-wire system for a pixel-formed TPEL device (3.5 × 2 cm), in which we used three different phosphor powders to fabricate three separated phosphor layers (Phosphors 1–3) aligned to the three different electrodes (Electrodes 1–3). Besides that, we also introduced a small extra Electrode 0 (0.5 × 2 cm, next to Electrode 1) connected to a protective neutral wire (PEN) line without any phosphor bladed but with polar bridge covering above it for independent control of each electrode. **b** CIE diagram of EL emissions corresponding to the three phosphor layers, with their (x,y) coordinates (0.188, 0.400); (0.464, 0.433) and (0.165, 0.266) for phosphors 1, 2, and 3, respectively. **c** Independent control of each electrode and the phosphor emission, simulating all eight different working states for the pixels. E0–E3 stand Electrodes 0–3. The used phosphors: GG45 (phosphor 1, green), GG14 (phosphor 2, orange), and GG65 (phosphor 3, blue). Commercial hydrogel was used as a polar bridge and PET-ITO was used as a substrate/electrode.

including double-insulation (without charge injection)[17,19,20,41–43] (including tandem devices[41]), single-insulation (charge injection from one electrode only)[18,44–47] and double-injection[39,48] OLEDs. To prove a wider applicability of driving EL devices by TP power source, we have fabricated and tested a series of TP-OLEDs with emission in red, green, cyan-blue, or blue regions (Fig. 6a, b). We used a single-insulation device structure as it is believed that it generally can lead to a higher performance compared to the structure with two dielectric layers (one on each side of the light-emitting layer). The devices were fabricated with a structure of ITO/dielectric layer/field-induced hole generation layer (FIHGL)/ emission layer (EML)/electron transporting layer (ETL)/electron injection layer (EIL)/aluminum (Al) (Fig. 6a). For the devices emitting different colors, the EML consisted of red (bis[2,4-dimethyl-6-[5-(2-methylpropyl)-2-quinolinyl-κN]phenyl-κC](2,4-pentanedionato-κO2,κO4) iridium, RD), green (bis[2-(2-pyridinyl-N)phenyl-C](2,4-pentanedionato-O2,O4)iridium(III), GD), or cyan-blue (bis[(4,6-di-fluoropheny)-pyridinato-C2',N](picolinato) iridium(III), FIrpic) phosphorescent dopants (Ir[III] complexes) or blue fluorescent dopant (N,N'-bis(2-methylphenyl)-N, N'-bis(6-tert-butyldibenzofuran-4-yl)pyrene-3,8-diamine, BD) in high bandgap semiconducting matrices (Fig. 6c). With this device structure, ITO separated from the functional layers by a dielectric layer represents an electric bridge (EB) (similar to PEB in inorganic TPEL), so only one type of charge carrier is directly injected from the external electrode (E = Al), whereas the other type of charge carriers is generated within the device.

We measured the TP voltage and corresponding current waveforms of the device and observed an obvious capacitive property in the TP-OLED with the currents to be ahead of the voltages (Fig. 7a). The relationships between the light output and the period of driving voltages for each phase clear show that the light emission occurs mainly in the negative half of the AC cycle for each phase, with substantially weaker (or almost no) light emission in the positive half-cycle (Fig. 7b).

In this case, the mechanism for the device operation is different from the previously discussed inorganic TPEL and the light emission occurs from the radiative recombination of excitons in the EML rather than the impact excitation of emitters by hot electrons[16]. The above observations can be explained as follows. TP-OLED represents three coplanar capacitors (Fig. 6a), which are turned into parallel plate out-of-phase capacitors connected by an EB (here we used transparent conductive ITO film as an EB), thus forming a set of parallel capacitors connected in a series. In the negative sinusoidal half-cycle of AC excitation in one of three capacitors E1(−)//EML//EB(+), the electrons are injected from E1 electrode onto lowest un-occupied molecular orbital (LUMO) energy level of ETL layer and drift toward EML (Supplementary Fig. 15, left). At the same time, the field-induced charge generation occurs in the FIHGL, consisting of a strong electron acceptor 1,4,5,8,9,11-hexaazatriphenylene hexacarbonitrile (HATCN) and strong electron donor 1-[4-(10-[1,1'-biphenyl]-4-yl-9-anthracenyl) phenyl]-2-ethyl-1H-benzimidazole (HTM), i.e., within a capacitor structure of ITO/dielectric layer/HATCN/HTM.

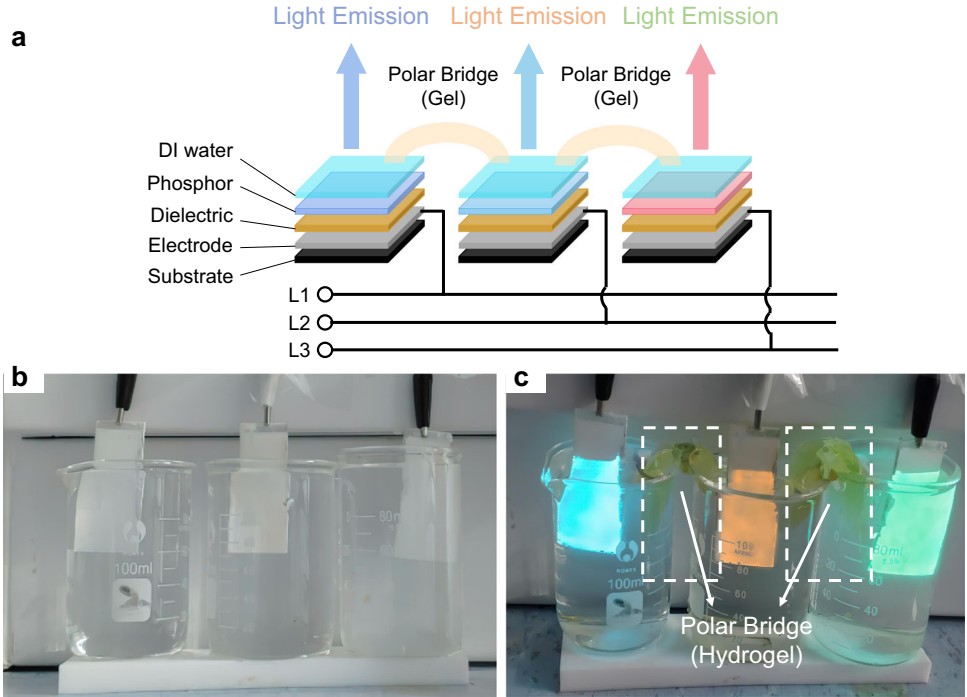

**Fig. 4 Polar bridge experiments using separated TPEL units. a** Schematic diagram of TPEL device separated into three parts, each driven by one single-phase AC supply of the three-phase electric power source, and connected together by a polar bridge. We used DI water as a polar bridge in each part, which are connected by hydrogel. **b**, **c** The photographs of three separate units, each connected to one single-phase AC supply of the three-phase electric power and placed into the beakers with DI water: **b** no bridges between the units (no light emission), **c** the units are connected by hydrogel bridges (each unit starts to emit the light from its phosphor layer). GG65, GG14, and GG45 were used as phosphors (from the left to the right) and glass-ITO was used as substrate/electrode.

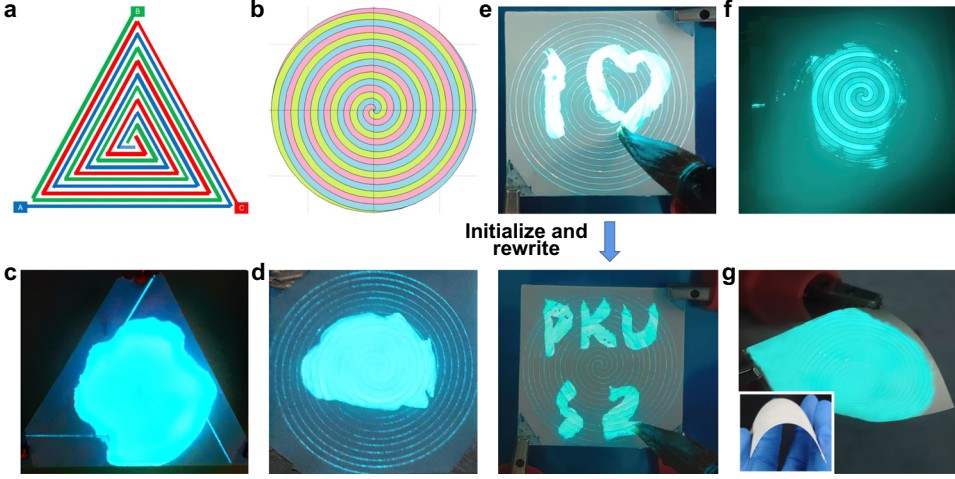

**Fig. 5 Multifunctional TPEL panels for interactive rewritable displays, optical-output sensors, and general lighting.** Schematic diagrams of triangle-type (**a**) and lollipop-type electrodes (**b**) for TPEL panels. Photographs of working triangle-type (**c**, 4 cm side) and lollipop-type (**d**, 5 cm diameter) TPEL panels with DI water as a polar bridge on the top of the devices. **e** Photographs of rewritable interactive display with written characters with DI water by a writing brush, erased by absorbent paper and re-written again. **f** Three-phase imbalance detection by remote optical alarm for simulative long-distance electric transmission lines using a TPEL panel. The three-phase voltages are 100, 110, and 120 V, respectively, at 60 Hz. **g** Flexibility of TPEL panels when using PET as a substrate.

Electrons from the highest occupied molecule orbital of HTM are tunneled into LUMO of HATCN to form the bound electron-hole pairs at the interface of donor–acceptor bilayer structure, which dissociate in the electric field, and the holes thus generated in HTM move toward EML layer under the electric field where they form excitons with electrons injected from E1 electrode and recombine with light emission. When the voltage is reversed in the next half-cycle to E1(+)//EML//EB(−), the less efficient hole injection from Al electrode occurs, which under an electric field move through the organic functional layers toward the dielectric layer to neutralize the negative charges in HATCN layer and near the dielectric surface, and refill the depleted FIHGL (Supplementary Fig. 15, right). As the two other E(+/−)//EML//EB(−/+) capacitors (with electrodes E2 and E3) work in the same manner of cycling but with a phase shift of 120°, this results in the current and the luminance of a TP-OLED to be more

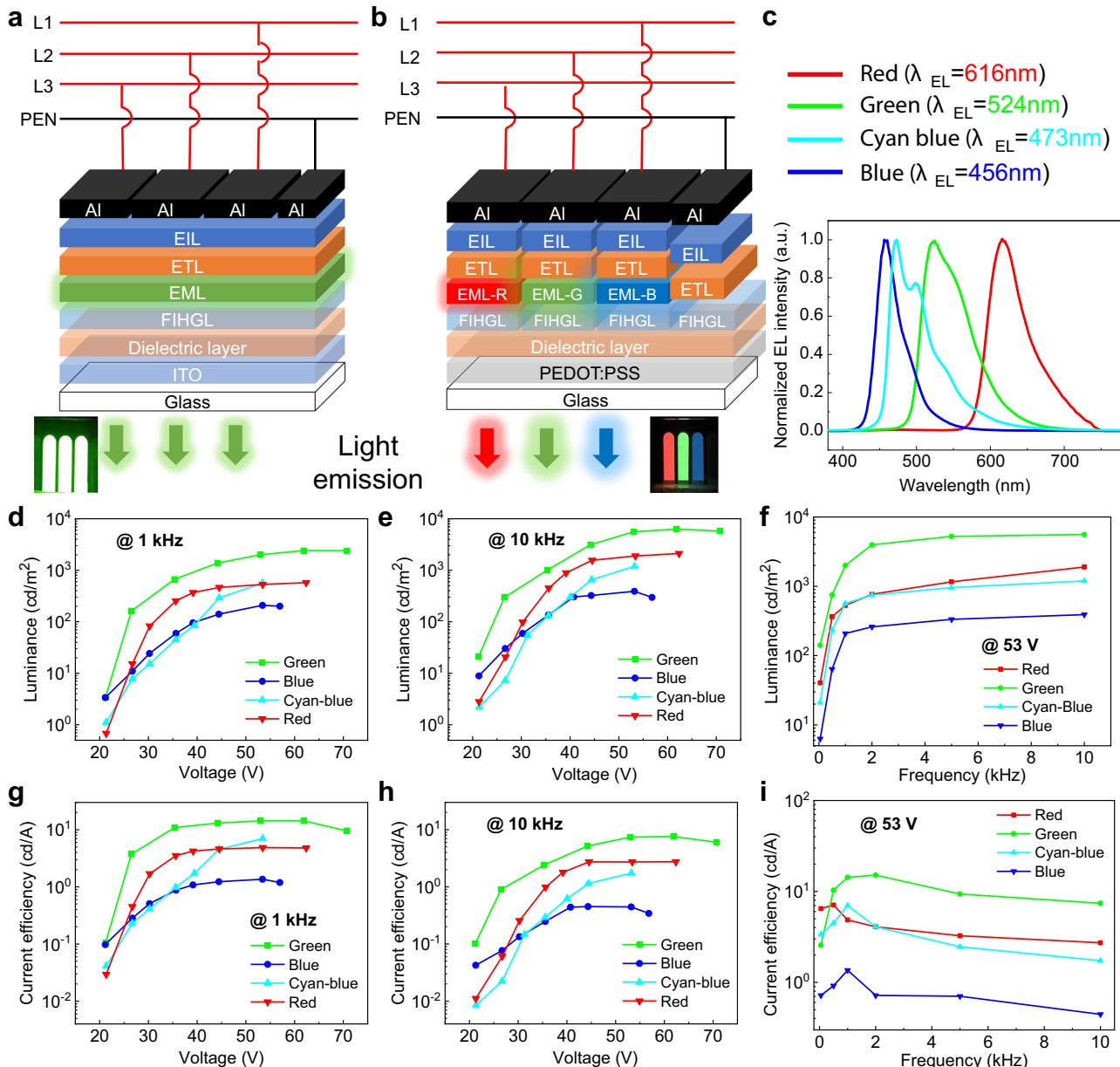

**Fig. 6 Schematic representation of TP-OLEDs and their characterization. a** Schematic structure of standard TP-OLEDs with a single EML. **b** Schematic structure of a pixel-formed TP-OLEDs with three uniplanar EMLs emitting red, green, and blue colors. The photographs for both devices are inserted in the bottom. **c** Normalized EL spectra of TP-OLEDs with different emitters. **d** Luminance and **g** current efficiency of standard TP-OLEDs as functions of voltage at the fixed frequency of 1 kHz. **e** Luminance and **h** current efficiency of standard TP-OLEDs as functions of voltage at the fixed frequency of 10 kHz. **f** Luminance and **i** current efficiency of standard TP-OLEDs as functions of AC frequency at the fixed voltage of 53 V. For green OLED, the ETL layer thickness is 30 nm (see "Methods" and Supplementary Information). Source data are provided as a Source Data file.

uniform within the time compared to a single-phase device (Fig. 7a, b).

We showed that the TP-OLEDs operate efficiently with all organic EMLs (Fig. 6d–i), achieving the performance, which is even higher than that of TPEL devices with inorganic phosphors. Thus, for the most efficient green TP-OLED, the maximum luminance and maximum current efficiency are $L_{max} = 6277$ cd/$m^2$ at 10 kHz (with CE = 7.6 cd/A) (Fig. 6e, h), and CE$_{max}$ = 14.3 cd/A at 1 kHz (with $L = 2403$ cd/$m^2$) (Fig. 6d, g), respectively. The effect of the frequency on performance is significant (Fig. 6f, i). At the frequencies of 50 Hz the luminance is quite low (e.g., for green OLED, $L = 180$ cd/$m^2$ at 71 V), but increased drastically with the frequency (see comparison of the performances vs. voltage at 50 Hz, 1, 5, and 10 kHz, Supplementary

Figs. 16 and 17). An increase of the luminance with the frequency at a fixed voltage of 53 V is shown in Fig. 6f. The device impedance Z can be represented as $Z = (Z'^2 + Z''^2)^{1/2}$, where $Z'$ is the resistive component and $Z'' = 1/2\pi fC$ is the capacitive reactance component relating to the device capacitance C at the frequency $f$[49]. As the frequency increases, $Z''$ decreases and the current density flowing through the device could be higher resulting in an increase the excitons concentration and thus leading to stronger luminance intensity. When it comes to a capacitor-like device, $Z''$ should be much bigger than $Z'$, so $Z \approx Z'' = 1/2\pi fC$ and the current density should be proportional to the frequency $f$ confirming the capacitive character of TP-OLEDs (Supplementary Fig. 18). The current efficiency vs. frequency relationships for all the devices (red, green, cyan-blue, and blue)

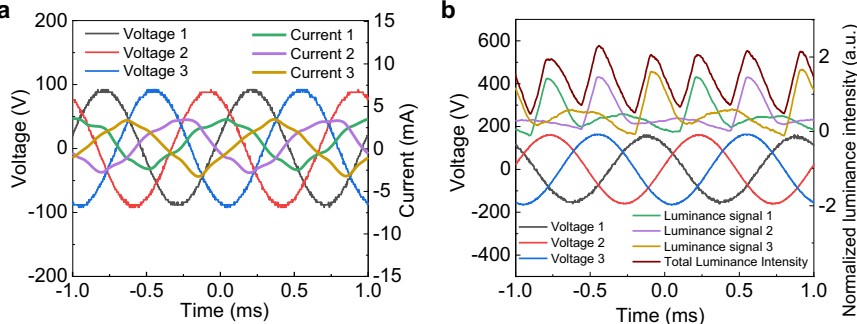

**Fig. 7 Phase characteristics of green TP-OLED. a** Oscilloscope signals of the three-phase voltage at $V_{rms} = 71$ V, 1 kHz and corresponding three-phase current oscillograms. **b** Relationship between the periods of three-phase driving voltage ($V_{rms} = 106$ V, 1 kHz) and the light output for each phase. Source data are provided as a Source Data file.

show peak values around 500–2000 Hz (Fig. 6i). The FIHGL provides free charge carriers only at a limited speed and amount, when the frequency is low and the injecting speed of electrons from Al electrode is slower than the maximum hole generation speed, in which case the current efficiency increases with the frequency. At high frequencies, the hole generation cannot catch up with the electrons injection resulting in the decrease of the current efficiency with the frequency.

We showed that TP-OLEDs can also be fabricated in a pixel form using a TP four-wire system with three different EMLs emitting in red, green, and blue (Fig. 6b). In this case, each electrode and thus luminous states of each pixel can be controlled independently (Supplementary Movie 6). To confirm the applicability of a wide range of materials as EB in TP-OLEDs, we fabricated pixel-formed TP-OLEDs on an ITO-free substrate, using a conducting polymer poly(3,4-ethylendioxythiophene)-poly(styrenesulfonate) (PEDOT : PSS) film as EB instead of ITO.

We also performed a preliminary optimization of the TP-OLED performance. With an increase of the thickness of ETL layer from 30 to 70 nm, the maxima of luminance, current efficiency, and power efficiency for the green TP-OLED were improved reaching the values of $L_{max} = 6601$ cd/m² at 10 kHz (with CE = 7.90 cd/A), $CE_{max} = 16.2$ cd/A at 1 kHz (with $L = 1403$ cd/m²), and $PE_{max} = 17.0$ lm/W at 1 kHz, respectively (Supplementary Fig. 19). We then compared the performance of the green TP-OLED (with an optimized ETL thickness of 70 nm) driven by a TP, single-phase, and sandwich electric systems (Supplementary Fig. 20 and 21). When driven by a TP electric power, the device showed the best performance, higher than in the case of driving the device by a single-phase ($L_{max} = 2019$ cd/m² at 10 kHz, $CE_{max} = 8.1$ cd/A at 1 kHz, $PE_{max} = 3.4$ lm/W at 1 kHz) or with a traditional sandwich driving system ($L_{max} = 4416$ cd/m² at 10 kHz, $CE_{max} = 9.6$ cd/A at 1 kHz, $PE_{max} = 8.9$ lm/W at 1 kHz) (Supplementary Fig. 21). As the TP-OLED configuration was the same in all three cases, with the same functional layers (EIL, ETL, EML, and FIHGL) and their thicknesses, we attribute the improved performance of the TP driving system to increased chances for carriers injection/generation and to more chances for the formation of excitons in the EML. These results demonstrate that OLEDs performance driven by a TP electric system is substantially improved comparing to the devices with a traditional sandwich configuration of SP power supply or single-phase driven AC-OLEDs.

## Disscussion

In the present work, we have reported a comprehensive method to drive EL devices by TP electric power and realized flexible, smartness, energy-saving, and multi-function devices. The structure and driving method of TPEL devices can be an inspiration for illumination and pixel formation. Our TPEL devices can be easily fabricated and scaled up using established technologies such as blade coating method without needs of expensive and work function matched transparent electrodes. We demonstrated pixel-formed TPEL devices and multifunctional TPEL panels applicable for interactive rewritable displays, optical-output sensors, as well as large-area solid-state lighting panels, displays, and information boards. The focus of the work was to demonstrate a new concept of EL devices driven by TP AC power. To demonstrate broad applicability of this concept, we extended it from inorganic phosphor-based TPEL devices to TP-driven OLEDs fabricated with red, green, and blue emitters, in both standard (single EML) and pixel-formed TP-OLED configurations. Although we were not specifically aimed to optimize TP-OLEDs configuration for achieving the highest device performance, the fabricated organic devices reached even higher luminance (up to 6601 cd/m², at 10 kHz) and current efficiency (up to 16.2 cd/A, at 1 kHz) as compared to inorganic TPEL devices. We also demonstrated that AC-OLED driven by a TP electrical system outperform those driven by a single-phase power. We believe that further developments using different luminescent materials (such as organic[17,50,51], polymer[52], perovskite[53], or quantum dots[54] EL materials), modification of the functional layers (such as retroreflective electrodes[55], dielectric layer modification[46], and electric field enhancing material-doped phosphor layer[18]), or advanced fabrication methods (such as photolithography) applying to such TPEL architectures should extend their potential applications and make the technology viable.

## Methods

**Materials.** Commercial PET-ITO or glass-ITO (both purchased from Hua Nan Xiang Cheng Technology, square resistance 6 Ω/cm²) were used as a substrate/electrode for device fabrications. In some experiments, tin foil or copper wire were used as electrodes (with no dielectric substrates). Mixture (1 : 1, by weight) of BaTiO₃ powder (diameter < 1 μm) and commercial binder (EL binder 026, Nanjing Collaborative Innovation Lighting) was used as a dielectric layer. Mixture (1 : 1, by weight) of commercial ZnS : Cu phosphor powders [GG45 (phosphor 1, green), GG14 (phosphor 2, orange), or GG65 (phosphor 3, blue), all purchased from Leuchtstoffwerk Breitungen GmbH, diameter < 30 μm] and commercial cyanoresin binder (EL binder 028, Nanjing Collaborative Innovation Lighting) was used to prepare light-emitting layer. DI water or commercial hydrogel (CICI jelly, Guangdong STRONG Group Co., Ltd) were used as PEBs. In some experiments, copper wire was used as a bridge electrode to connect DI water on each power line. Conductive polymer PEDOT : PSS was purchased from Xi'an Polymer Light Technology Corporation. Dielectric material poly(vinylidene fluoride-tri-fluoroethylene-chlorofluoroethylene) [P(VDF-TrFE-CFE)] was purchased from PiezoTech. GD was purchased from Xi'an Polymer Light Technology Corporation. FIrpic, 4,7-diphenyl-1,10-phenanthroline (BPhen), 2,6-bis[3-(9H-carbazol-9-yl) phenyl]pyridine (26DCzPPy), and 4,4',4"-tris(carbazol-9-yl)-triphenylamine (TCTA) were purchased from Guangdong Aglaia Optoelectronic Materials Co., Ltd. HATCN, HTM, 5-[3-(4,6-diphenyl-1,3,5-triazin-2-yl)phenyl]-5,7-dihydro-7,7-dimethylindeno[2,1-b]carbazole (H1), 5,7-dihydro-7,7-dimethyl-5-phenyl-2-

(9-phenyl-9H-carbazol-3-yl)indeno[2,1-b]carbazole (H2), 9-(1-naphthalenyl)-10-[4-(2-naphthalenyl)phenyl]anthracene (H3), RD, BD, 1-[4-[10-(1,1'-biphenyl-4-yl) anthracene-9-yl]phenyl]-2-ethyl-1H-benzimidazole (ETM), and 8-hydroxyquinolinolato-lithium (Liq) were donated from Nanjing USBT Co., Ltd.

**Materials used in TP-OLEDs fabrication**. The following materials have been used in fabrication of TP-OLEDs represented (their chemical structures are shown in Supplementary Fig. 14):

Dielectric layer: P(VDF-TrFE-CFE).

FIHGL: HATCN as electron acceptor and HTM or TCTA as electron donors.

EML hosts: H1, H2, or H3; 26DCzPPy.

EML dopants: RD for red EML, GD for green EML, BD for blue EML, and Flrpic for cyan-blue EML.

ETL: ETM or BPhen.

EIL: Liq.

**Fabrication of standard TPEL devices**. Commercial ITO films with glass or PET as substrates were used as electrodes (purchased from Hua Nan Xiang Cheng Technology, square resistance $6\,\Omega/cm^2$). A laser marking machine was used to print different electrode patterns. The substrates with ITO electrodes were sequentially cleaned by acetone, isopropanol, and DI water in an ultrasonic cleaner for 15 min each, dried in an oven, and then treated in a UV cleaner (320–500 nm) for 15 min before using in device fabrication. After pretreatment, a dielectric layer was made up of a 1 : 1 mixture (by weight) of $BaTiO_3$ powder (diameter < 1 μm) and commercial binder (EL binder 026, Nanjing Collaborative Innovation Lighting) using a blade coating method. Next, a phosphor layer made up of a 1 : 1 mixture (by weight) of commercial ZnS : Cu phosphor powder GG45 (phosphor 1, green), GG14 (phosphor 2, orange), or GG65 (phosphor 3, blue) (all purchased from Leuchtstoffwerk Breitungen GmbH, diameter < 30 μm) and commercial cyanoresin binder (EL binder 028, Nanjing Collaborative Innovation Lighting) was blade coated on the top of dielectrics. Both, the $BaTiO_3$/binder and ZnS : Cu/binder composites were thermally cured by a heating plate at 100 °C for 10 min after blade coating process.

**Fabrication of TP-OLEDs**. All the TP-OLEDs described in this paper were fabricated on glass substrates (1.5 cm × 1.5 cm). For standard TP-OLEDs, ITO conductive films pre-coated on glass substrates were used as electrode bridge (EB). ITO-coated glass substrates were sequentially cleaned with acetone, isopropanol, and DI water in an ultrasonic cleaning bath for 30 min each, dried with nitrogen gas flow, and then treated in a UV cleaner (320–500 nm) for 15 min. After this pretreatment, dielectric layer was deposited onto ITO from 100 mg/mL solution of P(VDF-TrFE-CFE) in N,N-dimethylformamide by spin-coating at 1000 r.p.m. for 30 s on air, followed by thermal annealing at 75 °C for 5 h. After that, organic layers (FIHGL, EML, ETL, and EIL) and Al electrodes were deposited sequentially by a vacuum thermal evaporation method at the base pressure of $5 \times 10^{-7}$ Torr. The thickness of the layers and the deposition rates during deposition were monitored by quartz crystal monitoring.

For the fabricated devices, the layers were deposited in the following order (the thickness of each layer is given in brackets):

Red-emitting TP-OLED: HATCN (10 nm)/HTM (30 nm)/65% H1 : 32.5% H2 : 2.5% RD (40 nm)/ETM (30 nm)/Liq (2.5 nm)/Al (100 nm).

Green-emitting TP-OLED: HATCN (10 nm)/HTM (30 nm)/48.8% H1 : 48.8% H2 : 2.4% GD (50 nm)/ETM (30 or 70 nm)/Liq (2.5 nm)/Al (100 nm).

Blue-emitting TP-OLED: HATCN (10 nm)/HTM (30 nm)/90.9% H3 : 9.1% BD (50 nm)/ETM (30 nm)/Liq (2.5 nm)/Al (100 nm).

Cyan-blue-emitting TP-OLED: HATCN (10 nm)/TCTA (40 nm)/90.9% 26DCzPPy : 9.1% FIrpic (50 nm)/BPhen (40 nm)/Liq (2.5 nm)/Al (100 nm).

For pixel-formed TP-OLEDs, PEDOT : PSS films were used as EB. After the pretreatment of glass substrate, PEDOT : PSS (4083) was spin-coated at 3000 r.p.m. for 30 s on air and then thermally annealed at 130 °C for 20 min. The next steps of deposition of dielectric layer, organic layers and Al electrodes are the same as for standard TP-OLEDs. The total effective luminous area for both standard TP-OLED and pixel-formed TP-OLED are 48 $mm^2$ (for pixel-formed TP-OLED, it is 16 $mm^2$ for green, red, and blue emitter, respectively).

**Operation methods**. TPEL devices in this work were driven by a specially designed TP power supply assembled in the lab, which generally includes two function generators (RIGOL DG4062, Rigol Technologies, Inc.), three voltage amplifiers (Aigtek ATA-2081, Aigtek Electronic Technology Co., Ltd), and a digital power meter (ZLG PA310, ZLG Technology Co., Ltd). This assembled TP power supply allowed to provide different TP voltages (0–800 $V_{p-p}$) and frequencies (from DC to AC 200 kHz) to drive TPEL devices in this work. For light emission in TPEL devices, an extra top-coated polar bridge above the phosphor layer is required. With using DI water as a polar bridge, TPEL devices showed stable light emission under 20–115 V voltages and 50–6000 Hz frequencies. A typical connection equivalent circuit diagram for connection of TPEL devices is shown in Supplementary Fig. 2b. Unless otherwise stated, we used DI water as a polar bridge and 1000 Hz, 80 V TP electric power to drive TPEL devices in this work.

**Characterization methods**. All optical characterizations were carried out in dark environment and exposed to the air. The luminance and chromaticity coordinates of the TPEL devices were measured by a luminance colorimeter (Topcon BM-7A; TOPCON TECHNOHOUSE CORP. ). We used DI water as a polar bridge, unless otherwise indicated, and assembled an extra optically transparent glass using 50 μm-thick double-sided tape (Hasegawa trytool, Hasegawa Corporation) as a tackiness agent and gasket above the top modulating layer during measurement, to ensure the stability of the polar bridge thickness. The power characteristics of TPEL devices were measured using a precision digital power meter (ZLG PA310; ZLG Technology Co., Ltd). Current and voltage waveforms were measured with a four-channel digital storage oscilloscope (Tektronix TDS 2024C; Tektronix, Inc.). The structures including surface and cross-section images were characterized by a field-emission SEM (ZEISS SUPRA 55, Carl Zeiss) and the elemental mapping showing the element distribution of the cross-section part of a TPEL device was measured by an energy dispersive spectrometer (Oxford X-Max 20; Oxford Instruments, plc.).

## Data availability

Data that support the finding of this study are available from the corresponding authors upon reasonable request. Source data are provided with this paper.

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

## Acknowledgements

This work was financially supported by the Key-Area Research and Development Program of Guangdong Province(2019B010924003), Shenzhen Engineering Laboratory (Shenzhen development and reform commission [2018]1410), Shenzhen Peacock Plan (KQTD2014062714543296), Shenzhen Science and Technology research grant (JCYJ20180302153514509, JCYJ20180302153406868), the Guangdong International Science Collaboration Base (2019A050505003), NSFC (Natural Science Foundation of China) No.61805004, and Shenzhen Key Laboratory of Organic Optoelectromagnetic Functional Materials (ZDSYS20140509094114164).

## Author contributions

H.M. conceived the idea, acquired funding for the project, and directed and supervised the research. J.J. designed the experiment, and fabricated and characterized devices. I.F.P., J.B., D.H., X.X., C.Z., T.W., and M.L. analyzed data and provided advice about the research. J.J., I.F.P., W.H., and H.M. prepared and discussed the manuscript and its revised version. All authors contributed to the analysis of the data and to the discussion of the manuscript.

## Competing interests

The authors declare no competing interests.
