## [Peer Review File · Nature Communications]

Editorial Note: This manuscript has been previously reviewed at another journal that is not operating a transparent peer review scheme. This document only contains reviewer comments and rebuttal letters for versions considered at Nature Communications .

Reviewers' comments:

Reviewer #1 (Remarks to the Author):

The response of the authors does not significantly address the concerns previously raised, and there are minimal changes to the paper. So the 3 main concerns remain :

1. that there is not a strong need for 3 phase operation of lighting
 2. the performances are very poor even at very high frequencies and even worse at 50 Hz. The author response that this was not the point of this study does not help - lighting needs to be efficient.
 3. The authors have already published AC devices which are the main innovation here.
- So the paper should be rejected.

Reviewer #4 (Remarks to the Author):

This paper entitled "Three-phase electric power driven electroluminescent devices" addresses two important problems in electroluminescent devices technology: the novel structure for EL devices and new driving method, using three phase AC current:

a) planar structure of electroluminescent (EL) device electrodes is introduced with polar electrode bridge (PEB) contrary to usual sandwich (or stacked structure) of conventional EL devices, usually having top transparent electrode

b) the new type of electrical driving method of EL devices by AC current is suggested and proved for three phase AC current driven structure introduced for the first time for EL devices.

Authors main claim is related with the advantages of the suggested for first time use of three phase AC current, as opposed to single phase AC driven EL devices. They call such devices as TPEL: three phase EL device for inorganic phosphorescent EL case, and TPOLED for three phase organic LED (i.e. OLED) devices. Authors have earlier already introduced a concept of a polar electrode bridge (PEB in ref 7 of this present manuscript) for inorganic EL devices which allows to use both active electrodes in a planar configuration while PEB (such as usual water or graphite powder or any conductive film) is used on the top of EL, instead of a traditional transparent electrode.

Now in this interesting new development the authors are further developing this concept of PEB, combining the PEB concept with three phase AC current (TP-AC) for driving this EL devices with PEB. Authors demonstrate this claims on two basic types of devices. Inorganic EL and Organic OLED. Moreover they demonstrate the several interesting advantages of this combination : PEB and TP-AC for simple pixilation technology and convenient 3 color pixilation using well known techniques.

In first 3 section the Inorganic phosphorous based TPEL devices are described in details based on their approach on earlier published paper on a "polar bridge.. (Ref.7).Indeed that is well written paper [7]and reader can understand the physics of processes from quite detailed description in Supplement to Ref 7. Both Claims are valid and Indeed authors present enough experimental data for demonstration that PEB permits to avoid the need of top transparent electrode, and in their pioneering TP-AC drive the parameters of TPEL device are exceeding those for single phase AC case (which they showed in Ref.7).

The real strength of this paper to my opinion is the completely new demonstration of application of PEB concept to TPOLED type device. They obtained here results that are not only conceptually new, but will be interesting to broader spectrum of researchers, since TPOLED is a new concept and it shows excellent performance (better than for inorganic TPEL) that can be practically used in industrial technology with numerous practical applications.

The weakness of paper is absence of good description of the physical processes taking place in TPOLED and lack of scientific discussion of this novel approach. Authors are not giving enough credit to earlier work on AC driven OLED. Moreover this earlier work is not cited appropriately: citations are made in a wrong place and without any details, just to a group (see page 15 just before acknowledgements) or in the SI part , but not in a relevant places in the main text. This may give a wrong impression that authors have pioneered an AC driven OLED structure with one dielectric layer coated electrode and only one injecting electrode, which they just use with PEB.

The following are some suggestions (which can be used to improve the paper).

Authors claim that while at each half phase the voltage is applied to base electrode and one of 3 planar electrodes, the PEB (that is a transparent liquid, like, water) allows the uniform distribution of strong electric field across emissive layer...and so on".

1. I suggest to improve this part by adding an energy level diagram for each pair of electrodes (e.g. in Supporting Information) for showing the physical processes, described above in more details. This will allow to a curious reader to better understand how the three phase operation allows to provide light emission by impact ionization and at next half period provide only the accumulation of charge (electrons or holes?) in trapping centers. Also it will help to compare with a single phase AC operation, described in Ref.7, which can be again reminded shortly. And here the proper citation of impact ionization processes in inorganic EL device will be useful to add to References.

2. In this regard the explanation of "holes being captured by the luminescent centers at first half period, while electrons travel to other side and accumulated at traps" might not be exactly correct for usual Inorganic EL case. Indeed at second half period, "when electrons are escaped from traps and accelerated causing impact ionization of Cu²⁺ impurities, with light emission", the description of holes moving in opposite direction and their role and position is probably missing ??

Usually in inorganic phosphorous EL devices the electrons generate light only via impact ionization of impurities, such as Cu²⁺. And holes are not needed in EL devices, since electron-hole recombination is not taking place, (like in OLEDs, where namely electron and hole recombination is key EL reason).

This part should be either corrected, or better explained with a citation to related papers.

Certainly the strongest part of this paper is extension of PEB concept to TPOLED. This is a part, which will attract most attention despite great results presented the explanation of operation physics is not done.

In this section, even more improvements are suggested:

3. First of all the extension of a polar bridge concept to ITO electrode (or to PEDOT-PSS) that are both standard transparent top electrode of most EL devices needs more detailed explanation. How ITO works as PEB in OLED, as compared to water dipoles in previously described PEG in TPEL? Is it better or worse than dipoles of water for possible PEB operation in OLED?

4. Thus the use of ITO is neither very attractive, nor is it justified, and it does not sound very exciting, since it is a usual bottom electrode. Not a top transparent, which authors claim to improve by eliminating it and using PEB. Now in TPOLED example ITO is PEB itself. It is not clear why it is better to use ITO in the bottom (as in all the usual devices), instead of using non transparent (e.g. Ag electrode) in the bottom and then use PEB, such as water or other polar liquid on top of organic OLED top layer which is protected by dielectric insulating layer. It would be more impressive to show that the 4 pixel structure in which one of 4 is a base electrode, and 3 others are active electrodes for 3 color pixels, can work without any ITO or PEDOT-PSS, but water type PEB.

At least authors should discuss such geometry and show the possibility to create a completely transparent TPOLED if pixelated ITO is used on bottom and water type PEB from top (which is main advantage of this paper).

Most probably this is not an effective device, since water type polar bridge will not work well on organic OLED as compared to Inorganic TOEL pixel? This needs explanation.

5. Moreover when authors move to next concept of TP AC for OLEDs, they do not describe why they have chosen the architecture of one insulating layer OLED. In fact the different types of AC driven OLEDs (which have already a long history, since 1996 [1-2, see list below] is not properly cited. Authors can shortly describe why they have chosen a configuration of OLED with one injecting electrode and other electrode is coated with insulator. This needs clarification, since operation of TPOLED is quite different from inorganic TPEL:

However, the authors do not give enough credit to pioneering work on single layer AC-OLED [1-3] and other important papers here (list of some of key papers is given below [4-8] in which these initial devices rely either on charge injection from one or even both electrodes.

And AC-driven OLED usually show poor performance when operated in a full insulating, capacitively coupled mode, i.e., in a configuration where two insulators prevent charge injection from both electrodes.

Authors should shortly describe the very different physical processes for inorganic EL versus OLED, which still allowed them to use the advantages of 3 phase versus 1 phase AC for OLED with the lateral position of two electrodes, bridged by only one capacitive coupling via polar PEB bridge.

And this is their main claim. All other novelties are related with technological advantages of lateral electrodes.

Some suggested new References for possible citations (maybe partially only on authors choice):

1. Y. Z. Wang, D. D. Gebler, L. B. Lin, J. W. Blatchford, S. W. Jessen, H. L. Wang, and A. J. Epstein, Appl. Phys. Lett. 68, 894 (1996).
2. T. Tsutsui, S.-B. Lee, and K. Fujita, Appl. Phys. Lett. 85, 2382 (2004).
3. J. Sung, Y. S. Choi, S. J. Kang, S. H. Cho, T.-W. Lee, and C. Park, Nano Lett. 11, 966 (2011).
4. Y. Chen, Y. Xia, G. M. Smith, Y. Gu, C. Yang, and D. L. Carroll, Appl. Phys. Lett. 102, 013307 (2013).
5. M. Fréchet, A. Perumal, T. Schwab, C. Fuchs, K. Leo, and M. C. Gather, Phys. Status Solidi A 210, 2439 (2013).
6. A. Perumal, M. Fréchet, S. Gorantla, T. Gemming, B. Lüssem, J. Eckert, and K. Leo, Adv. Funct. Mater. 22, 210 (2012).
7. A. Perumal, B. Lüssem, and K. Leo, Appl. Phys. Lett. 100, 103307 (2012).
8. A. Perumal, B. Lüssem, and K. Leo, Org. Electron. 13, 1589 (2012).

Reviewer #5 (Remarks to the Author):

In this work, the author demonstrated that light-emitting devices can be directly driven by TP phase and showed the advantages of this approach. And most of the issues suggested by the referees have been addressed in the revised manuscript. I think this manuscript can be accepted after revisions as follows.

(1) The performances of these TP AC devices are relatively poor. The current efficiencies are lower than the corresponding conventional devices and the power efficiencies are even lower due to the high driving voltages, both of which would eliminate the advantages of the TP AC devices. I am wondering if the authors could provide devices with higher performance or not? At least they should explain the main factors that limit the performance of the current design.

(2) Also, I agree with reviewer #3, the voltage-time-EL for the three electrodes independently should be given and more expression and evidence should be given on the operating mechanism.

Our replies on the reviewers' comments are given below.

Reviewer #1

The response of the authors does not significantly address the concerns previously raised, and there are minimal changes to the paper. So the 3 main concerns remain:

1. that there is not a strong need for 3 phase operation of lighting

We thank the reviewer for different opinion, and we respect the personal opinion of the reviewer, but our view on the subject is different. The reviewer may come to this conclusion for poor performance of AC LEDs compared to DC devices. But our vision is that any novel conceptual approach worth attention, opening new doors for developments (materials, devices, technologies) even if they are not ready at the moment for immediate implication. We believe that we clearly argued as to why TP driving of EL devices might become vital technology and occupy its own niche in EL device applications.

Reviewer #4 also considered the proposed approach as conceptually new and interesting. Thus, different opinions among the peers only reflect the diversity of views, but not a weakness of this work.

2. the performances are very poor even at very high frequencies and even worse at 50 Hz. The author response that this was not the point of this study does not help - lighting needs to be efficient.

For inorganic TPEL, we used known commercial phosphors, which already used in single-phase ACEL commercial products in the past decades. We agree that their performance is low, but they occupy their own niche where the brightness is not a critical issue. For some applications we demonstrated (pixel-formed TPEL, rewritable panel, remoted safety phase control, see e.g. video, Fig. 5) the luminance is at the acceptable level.

Also, we think that no one can expect that changing the driving scheme from SP to TP would drastically change the performance from the same material. We tried to demonstrate that EL device can be driven by TP (the approach which was not demonstrated before), and apart of general advantage of direct TP driving the devices, an improvement toward somewhat increased efficiency and more uniform light emission with a time can be realized.

We extended the concept to TP-OLED and in this case, we demonstrated that rather high performance of devices can be achieved (initially reported 5760 cd/m² was increased to 6601 cd/m² in the revision).

On our opinion, achieving highest performance is a general challenge of existing AC driven devices (used materials, device structure etc) and not an issue of TP driving approach.

3. The authors have already published AC devices which are the main innovation here.

The main innovation of this manuscript is not just demonstration of AC-driven EL devices, which were studied before in many publications (and we put many references on that, e.g. [17-23, 33-43] including two review articles [15,16]), but the proof of the possibility of using three phase AC power supply for direct driving of both inorganic EL devices and organic OLEDs and the benefits arising such driving scheme of light-emitting devices.

The three-phase AC power scheme was never used before for driving electroluminescent devices.

Reviewer #4

This paper entitled "Three-phase electric power driven electroluminescent devices" addresses two important problems in electroluminescent devices technology: the novel structure for EL devices and new driving method, using three phase AC current:

a) planar structure of electroluminescent (EL) device electrodes is introduced with polar electrode bridge (PEB) contrary to usual sandwich (or stacked structure) of conventional EL devices, usually having top transparent electrode

b) the new type of electrical driving method of EL devices by AC current is suggested and proved for three phase AC current driven structure introduced for the first time for EL devices.

Authors main claim is related with the advantages of the suggested for first time use of three phase AC current, as opposed to single phase AC driven EL devices

Now in this interesting new development the authors are further developing this concept of PEB, combining the PEB concept with three phase AC current (TP-AC) for driving this EL devices with PEB. Authors demonstrate this claims on two basic types of devices. Inorganic EL and Organic OLED. Moreover, they demonstrate the several interesting advantages of this combination: PEB and TP-AC for simple pixilation technology and convenient 3 color pixilation using well known techniques. In first 3 section the Inorganic phosphorous based TPEL devices are described in details based on their approach on earlier published paper on a “polar bridge. (Ref.7). Indeed that is well written paper [7] and reader can understand the physics of processes from quite detailed description in Supplement to Ref 7. Both Claims are valid and Indeed authors present enough experimental data for demonstration that PEB permits to avoid the need of top transparent electrode, and in their pioneering TP-AC drive the parameters of TPEL device are exceeding those for single phase AC case (which they showed in Ref.7).

The real strength of this paper to my opinion is the completely new demonstration of application of PEB concept to TPOLED type device. They obtained here results that are not only conceptually new, but will be interesting to broader spectrum of researchers, since TPOLED is a new concept and it shows excellent performance (better than for inorganic TPEL) that can be practically used in industrial technology with numerous practical applications.

We thank the reviewer for these comments supporting the novelty of our work and the significance of our claims in the manuscript.

The weakness of paper is absence of good description of the physical processes taking place in TPOLED and lack of scientific discussion of this novel approach. Authors are not giving enough credit to earlier work on AC driven OLED. Moreover this earlier work is not cited appropriately: citations are made in a wrong place and without any details, just to a group (see page 15 just before acknowledgements) or in the SI part, but not in a relevant places in the main text. This may give a wrong impression that authors have pioneered an AC driven OLED structure with one dielectric layer coated electrode and only one injecting electrode, which they just use with PEB.

In the introduction (page 2), we mentioned previous works on AC driven OLED devices and cited two review papers on the subject (“Recently, alternating current (AC) driven electroluminescent (EL) devices have attracted increased attention and are regarded as promising alternatives to traditional DC-driven EL devices [15,16]”). Several more papers had been cited in further text in the introduction (ref [17-20,23]). With that (and considering that the titles of these paper are available in the list of references), we do not think that the readers can be confused that we pioneered on using AC driven OLED concept.

Also, in line with the reviewer suggestions, we expanded the TP-OLEDs section in the Results and Discussion, with additional reference on pioneering works, exploiting different structures of TP-OLED devices and the mechanisms of their operation [ref. [39–43, 47–49].

Regarding the references given in the end of the manuscript which we had in the initial submission (and left in the revision) these were (are) used to underline the possible directions in further developments in TP-OLEDs which are crucial for improving the device performance, to make the technology viable.

I. I suggest to improve this part by adding an energy level diagram for each pair of electrodes (e.g. in Supporting Information) for showing the physical processes, described above in more details. This will allow to a curious reader to better understand how the three phase operation allows to provide light emission by impact ionization and at next half period provide only the accumulation of charge (electrons or holes?) in trapping centers. Also it will help to compare with a single phase AC operation, described in Ref.7, which can be again reminded shortly. And here the proper citation of impact ionization processes in inorganic EL device will be useful to add to References.

We have amended the text (pages 7-9), better explaining the mechanism of operation of EL devices driven by three-phase electrical source basing on (and compared to) systems driven by a single-phase AC. As exact mechanism of powder ZnS-type semiconductors is still under debates in the literature (e.g. with general view on charge generation/separation and recombination in capacitive devices during the reverse cycles, the places where the events occurs are questionable), we just focused on consideration what happens when TP power with a phase shift is applied to the capacitors.

2. In this regard the explanation of “holes being captured by the luminescent centers at first half period, while electrons travel to other side and accumulated at traps” might not be exactly correct for usual Inorganic EL case. Indeed at second half period, “when electrons are escaped from traps and accelerated causing impact ionization of Cu²⁺ impurities, with light emission”, the description of holes moving in opposite direction and their role and position is probably missing ?? Usually in inorganic phosphorous EL devices the electrons generate light only via impact ionization of impurities, such as Cu²⁺. And holes are not needed in EL devices, since electron-hole recombination is not taking place, (like in OLEDs, where namely electron and hole recombination is key EL reason). This part should be either corrected, or better explained with a citation to related papers.

We are sorry about unclear explanation of functioning and light emission mechanism from TPEL/ACEL and have revised the text accordingly. In fact, the mechanism from powder phosphor is still not fully understood, with debates and controversial experimental data. An excitation by hot electrons is more likely occurs with rare-metal doped ZnS, whereas for ZnS:Cu, the recombination mechanism is more acceptable, while the intimate details are still disputable.

We added more detailed explanation with corresponding literature references on that to make clearer how the devices work when they are driven by a three-phase electrical source. It is not very different from the scenario of AC driving by a single-phase power, except that the difference in potentials, when a single capacitor is considered, is due to difference in phases between the electrodes (instead of just difference in a voltage between two electrodes in SP case).

Certainly the strongest part of this paper is extension of PEB concept to TPOLED. This is a part, which will attract most attention despite great results presented the explanation of operation physics is not done. In this section, even more improvements are suggested:

3. First of all the extension of a polar bridge concept to ITO electrode (or to PEDOT-PSS) that are both standard transparent top electrode of most EL devices needs more detailed explanation. How ITO works as PEB in OLED, as compared to water dipoles in previously described PEG in TPEL? Is it better or worse than dipoles of water for possible PEB operation in OLED?

In the revision, we have substantially extended the section “Fabrication and characterization of TP-OLEDs” (pages 14-18) by adding such explanation of our choice to use single-insulation configuration of TP-OLEDs and using either ITO or PEDOT-PSS transparent electrodes.

Particularly, we aimed to demonstrate possibility to achieve higher performance and stability of the devices driven by TP, whereas exploiting PEB would require more extended studies toward applicability of organic materials that would not degrade in contacts with water or gel PEB. We consider more detailed studies of TP-OLEDs with polar bridges in our current studies, which will be published separately in the future.

Regarding the operation of ITO instead of polar PEB, we don't see problems here as we showed previously that conductive bridges work equally well (for either organic and inorganic materials). In our previous publication (ref. [7] in the manuscript) we showed that other sufficiently polar solvents instead of distilled water work well (e.g. ethanol, ethylene glycol or aqueous NaCl, so the conductivity is not determining factor (when it is above the certain level)). Metallic wire bridges also worked well in TPEL, except that the materials are not transparent; but the emission was clear observed around the wire contacts when they used as PEB.

4. Thus the use of ITO is neither very attractive, nor is it justified, and it does not sound very exciting, since it is a usual bottom electrode. Not a top transparent, which authors claim to improve by eliminating it and using PEB. Now in TPOLED example ITO is PEB itself. In is not clear why it is better to use ITO in the bottom (as in all the usual devices), instead of using non transparent (e.g. Ag electrode) in the bottom and then use PEB, such as water or other polar liquid on top of organic OLED top layer which is protected by dielectric insulating layer. It would be more impressive to show that the 4 pixel structure in which one of 4 is a base electrode, and 3 others are active electrodes for 3 color pixels, can work without any ITO or PEDOT-PSS, but water type PEB. At least authors should discuss such geometry and show the possibility to create a completely transparent TPOLED if pixelated ITO is used on bottom and water type PEB from top (which is main advantage of this paper).

We agree with the reviewer that TP-OLED in a configuration with using PEB instead of ITO bottom electrode would be an attractive and challenging option. However, this would require more studies toward device stability (used organic materials) and performance, and no one can fit all possible/desired studies in a single paper. We selected to demonstrate the applicability of TP for driving not only inorganic phosphor ACEL but also for OLEDs and for this used configuration which would demonstrate acceptably high efficiency (as requested by the reviewers).

Most probably this is not an effective device, since water type polar bridge will not work well on organic OLED as compared to Inorganic TOEL pixel? This needs explanation.

Yes, the reviewer is right, – water might be not suitable for OLED (at least, in used configuration and wish used materials), degrading the devices. This was one of the reasons as to why we decided to use ITO or PEDOT:PSS to demonstrate applicability of TP driving to OLEDs. We have added short explanation to the text discussing the selection of our TP-OLED configuration.

5. Moreover when authors move to next concept of TP AC for OLEDs, they do not describe why they have chosen the architecture of one insulating layer OLED. In fact the different types of AC driven OLEDs (which have already a long history, since 1996 [1-2, see list below] is not properly cited. Authors can shortly described why they have chosen a configuration of OLED with one injecting electrode and other electrode is coated with insulator. This need clarification, since operation of TPOLED is quite different from inorganic TPEL.

We have added more citations, as suggested by the reviewer, and have explained the choice of single-insulation/single-injection configuration for TP-OLEDs (to the section “Fabrication and characterization of TP-OLEDs.”).

However, the authors do not give enough credit to pioneering work on single layer AC-OLED [1-3] and other important papers here (list of some of key papers is given below [4-8] in which these initial devices rely either on charge injection from one or even both electrodes.

We have added some references from the suggested to the manuscript (ref. [39], [41-44] and [49] in the revision, in addition to already cited ref. 17,18]) with some more discussion on the subject.

And AC-driven OLED usually show poor performance when operated in a full insulating, capacitively coupled mode, i.e., in a configuration where two insulators prevent charge injection from both electrodes.

We agreed with the reviewer. This is why we have chosen a single-insulating configuration, to show better performance from our TP-OLEDs.

Authors should shortly describe the very different physical processes for inorganic EL versus OLED, which still allowed them to use the advantages of 3 phase versus 1 phase AC for OLED with the lateral position of two electrodes, bridged by only one capacitive coupling via polar PEB bridge. And this is their main claim. All other novelties are related with technological advantages of lateral electrodes.

We have added more experiments to show advantages of TP-OLEDs over SP-driven devices (Fig. 7 and S14) and discussed the difference in mechanism of operation of TP-OLEDs (exciton recombination) vs inorganic TPEL (hot electrons impact). Fig. S12 has been added to demonstrate the scheme of TP-OLED operation with a text in the section “Fabrication and characterization of TP-OLEDs.”

Some suggested new References for possible citations (maybe partially only on authors choice):

1. Y. Z. Wang, D. D. Gebler, L. B. Lin, J. W. Blatchford, S. W. Jessen, H. L. Wang, and A. J. Epstein, *Appl. Phys. Lett.* 68, 894 (1996).
2. T. Tsutsui, S.-B. Lee, and K. Fujita, *Appl. Phys. Lett.* 85, 2382 (2004).
3. J. Sung, Y. S. Choi, S. J. Kang, S. H. Cho, T.-W. Lee, and C. Park, *Nano Lett.* 11, 966 (2011).
4. Y. Chen, Y. Xia, G. M. Smith, Y. Gu, C. Yang, and D. L. Carroll, *Appl. Phys. Lett.* 102, 013307 (2013).
5. M. Fröbel, A. Perumal, T. Schwab, C. Fuchs, K. Leo, and M. C. Gather, *Phys. Status Solidi A* 210, 2439 (2013).
6. A. Perumal, M. Fröbel, S. Gorantla, T. Gemming, B. Lüssem, J. Eckert, and K. Leo, *Adv. Funct. Mater.* 22, 210 (2012).

7. A. Perumal, B. Lüsse, and K. Leo, *Appl. Phys. Lett.* 100, 103307 (2012).

8. A. Perumal, B. Lüsse, and K. Leo, *Org. Electron.* 13, 1589 (2012).

We thank the reviewer for the suggestion to consider the above references on early pioneering and key works in the field. Actually, we cited two review articles on the subject (ref. [15,16] in the manuscript), which contain some of the above references (ref. 3-8). Also, ref. 3 and 6 from the above list were already cited in our initial submission (as ref. [18] and [17] in the manuscript, respectively). As suggested by the reviewer, we have added some more citations from the above list (ref. [39], [41-44], [49] in the revision).

Reviewer #5

In this work, the author demonstrated that light-emitting devices can be directly driven by TP phase and showed the advantages of this approaches. And most of the issues suggested by the referees have been addressed in the revised manuscript.

I think this manuscript can be accepted after revisions as follows.

(1) The performances of these TP AC devices are relatively poor. The current efficiencies are lower than the corresponding conventional devices and the power efficiencies are even lower due to the high driving voltages, both of which would eliminating the advantages of the TP AC devices. I am wondering if the authors could provide devices with higher performance or not? At least they should explain the main factors that limit the performance of the current design.

One should take into account that we did not use newly development materials but used known commercial materials, and we did not perform an optimization of device structures, the layers which can improve the performance etc. Our aim was to demonstrate that three-phase driven systems can work as principally novel approach, and we also showed that in general the performance of TP driven systems is better than classical SP AC EL system (toward somewhat better performance characteristic and uniformity of the emission over the time).

We think that the demonstrated performance of TP-OLED was high enough and with additionally requested experiments we even improved it reaching luminance of ca. 6600 cd/m² (that exceed well say the required brightness for displays or even large panels, although might be not very high if to consider general solid state lighting applications).

We agree that the performance of inorganic TPEL is relatively poor. This is because we used standard commercial phosphors as emitting materials with their characteristic performance, although again, the TP driven devices showed improved performance and more uniform light emission compared to single-phase analogs in the same configuration. SP ACEL based on powder ZnS-type phosphors have already been commercialized products and occupied their own niche of applications (e.g. low resolution information screens, control panels, safety displays etc.).

Achieving brighter emission and higher power efficiency would require using other materials and optimization of the devices structures (e.g. thin films inorganic ACEL materials), which is out of the aim of this work.

(2) Also, I agree with reviewer #3, the voltage-time-EL for the three electrodes independently should be given and more expression and evidence should be given on the operating mechanism.

Thank you for the suggestion. We have added experiments on voltage-time dependences of the current and the luminance for each phase of TP-OLEDs (Fig. 7) and discussed the mechanism of their operation for unsymmetrical single-insulation devices used in our work (section "Fabrication and characterization of TP-OLEDs.").

Reviewers' Comments:

Reviewer #1:

Remarks to the Author:

There is a defect in the paper which I raised in an earlier round of review that surprisingly still has not been corrected. This is that in figure 6 luminance-voltage is shown for 10 kHz and current efficiency-voltage at 1 kHz. I cannot understand why the authors present these at different frequencies. The reader needs to see these data at the same frequency so they can be confident in the validity of the measurements presented. The figure must be changed.

In fact, far from fixing this problem, the authors have added to it in figure S16 by comparing current efficiency-voltage at 1 kHz with luminance-voltage at 10 kHz. This is really poor – a good undergraduate project student would realise that to compare the data they should be at the same frequency.

Also the revised paper gives the impression that high efficiency and luminance are achieved simultaneously. To make the performance clear, the current efficiency should be stated at a brightness (e.g. 16.2 Cd/A at a brightness of xxx Cd/m²) and the frequency of operation must be clearly stated.

The authors also need to state clearly the efficiency achieved at 60 Hz because that is the frequency of 3 phase power, not 1 kHz and not 10 kHz. They argue that further work is needed, but we do at least need to know what the starting point is.

The authors argue that efficiency was never the point of the paper, and that low efficiency is acceptable in some niche applications. There are indeed such niche applications, but they are very small and not nearly enough to justify the use of 3-phase power. The devices the authors have operating at 1-2 kHz have power efficiency around a factor 50 lower than a state of the art OLED. But at 60 Hz the efficiency may be much worse. I assume it is much worse because the authors seem to be hiding the data.

Reviewer #4:

Remarks to the Author:

Authors have done significant modifications following my comments and suggestions. this paper is now ready for publication and I am sure the readers of this interesting manuscript now will be satisfied by details and will be able to understand the concepts better.

Reviewer #5:

Remarks to the Author:

The revised manuscript is acceptable in my opinion.

Our replies on the reviewers' comments are given below.

We thank all the reviewers for the time they spent on reviewing our manuscript and useful suggestions on improving its quality.

Reviewer #1

There is a defect in the paper which I raised in an earlier round of review that surprisingly still has not been corrected. This is that in figure 6 luminance-voltage is shown for 10 kHz and current efficiency-voltage at 1 kHz. I cannot understand why the authors present these at different frequencies. The reader needs to see these data at the same frequency so they can be confident in the validity of the measurements presented.

The figure must be changed.

We thank the reviewer for the suggestions. The reason for showing “luminance – voltage” (at 10 kHz) and “current efficiency – voltage” (at 1 kHz) at different frequencies was based on the highest efficiencies of the devices driven at 53 V (Fig. 6F,I). Our logic was to demonstrate what is the highest performance is achievable for a given parameter. The luminance was increased with the frequency showing highest value at *ca.* 10 kHz (Fig. 6F) and we selected 10 kHz frequency to look at the dependence of this parameter on a voltage. Current efficiency dependences reached maxima at *ca.* 1 kHz and dropped down after that (Fig. 6I). Therefore, the dependences on voltage were shown for 1 kHz.

Considering that this might be confusing for readers, we have followed the suggestion from the reviewer and added to the Fig. 6 two more figures showing the “ $L - V$ ” dependence at 1 kHz (Fig. 6D) and the “ $CE - V$ ” dependence at 10 kHz (Fig. 6H). We also added to the Supplementary more figures showing the “ $L - V$ ”, “ $CE - V$ ” and “ $J - V$ ” dependences at different frequencies (50 Hz, 1 kHz, 5 kHz and 10 kHz), so readers can easily compare the results (Fig. S16 and S17 in the SI). We have amended the text in the manuscript explaining that and to avoid misunderstanding.

In fact, far from fixing this problem, the authors have added to it in figure S16 by comparing current efficiency-voltage at 1 kHz with luminance-voltage at 10 kHz. This is really poor – a good undergraduate project student would realise that to compare the data they should be at the same frequency.

(Due to new added figures, Fig. S16 from previous manuscript version is now Fig. S19)

The reason of the choice of the frequencies at which “ $L - V$ ” and “ $CE - V$ ” were shown has already been explained above. Fig. S16 from previous manuscript version showed performance of the devices at two thicknesses of the electron transporting layer (30 and 70 nm), which we only used as intermediate data to choose the device structure (toward ETL thickness) for further studies.

According to the reviewer's request, we have added additional graphs, so both Fig. S19 and Fig. S21 show now “ $L - V$ ” and “ $CE - V$ ” at both 1 and 10 kHz frequencies. We have also added to both figures the graphs of power efficiencies at 1 and 10 kHz, for comparison.

Also the revised paper gives the impression that high efficiency and luminance are achieved simultaneously. To make the performance clear, the current efficiency should be stated at a brightness (e.g. 16.2 Cd/A at a brightness of xxx Cd/m²) and the frequency of operation must be clearly stated.

We have amended the text, clearly stating at which frequencies the parameters have been measured to avoid misunderstanding that the maxima of the parameters are achieved simultaneously. The maxima for each parameter mentioned in the text (L_{\max} , CE_{\max} , PE_{\max}) are achieved at different voltage (and frequencies). We have also indicated in the text complete information for each case (i.e. what was the brightness at CE_{\max} and current efficiency at L_{\max}).

The authors also need to state clearly the efficiency achieved at 60 Hz because that is the frequency of 3 phase power, not 1 kHz and not 10 kHz. They argue that further work is needed, but we do at least need to know what the starting point is.

The authors argue that efficiency was never the point of the paper, and that low efficiency is acceptable in some niche applications. There are indeed such niche applications, but they are very small and not nearly enough to justify the use of 3-phase power. The devices the authors have operating at 1-2 kHz have power efficiency around a factor 50 lower than a state of the art OLED. But at 60 Hz the efficiency may be much worse. I assume it is much worse because the authors seem to be hiding the data.

We have added Fig. S16 and S17 showing a comparison of the characteristics of the device (L , CE , PE) at different frequencies (50 Hz, 1, 5 and 10 kHz). Particularly, Fig. S16(left) shows “ $L - V$ ”, “ $CE - V$ ” and “ $J - V$ ” characteristics of TP-OLED at 50 Hz as “starting point”. We also mentioned, as suggested, the luminance at 50 Hz of the green TP-OLED in the text.

The low efficiency at 50–60 Hz frequencies is mainly an issue of materials. Developing of new materials was not a subject of this manuscript, as we intentionally used known, commercially available materials. And secondly, the state-of-the-art development of AC OLED technology is such, as known, that their device performance is lower than that of DC OLED. Nevertheless, added Fig. S21 clear shows that the performance of AC OLED devices can be improved when driven by three-phase power (as compared to single-phase or sandwich electric systems).

Reviewer #4

Authors have done significant modifications following my comments and suggestions. this paper is now ready for publication and I am sure the readers of this interesting manuscript now will be satisfied by details and will be able to understand the concepts better.

We thank the reviewer for this positive comment.

Reviewer #5

The revised manuscript is acceptable in my opinion.

We thank the reviewer for this positive comment.

REVIEWERS' COMMENTS

Reviewer #1 (Remarks to the Author):

The authors have provided additional information that addresses the most serious problems of the data presented. By providing current efficiency, light output etc. for each frequency the reader can now get a much better understanding of the performance that has been achieved so far. Quoting efficiency with a brightness is also a marked improvement and for high frequency drive, reasonable efficiency is achieved at useful brightness. Efficiency at mains frequency is lower, but at least provides a starting point.

Whilst I still have great difficulty envisaging real use of three phase driving of such devices, as the most serious concerns have been addressed, I do not object to publication of the paper.

Reviewer #4 (Remarks to the Author):

AUTHORS have accounted the comments of reviewer 1. I am satisfied with added new Figs on luminance and current efficiencies at 1 KHz and 10 KHz.

And with all other additions and changes. Now manuscript can be accepted for publication.

Our replies on the reviewers' comments are given below.

We thank all the reviewers for the time they spent on reviewing our manuscript.

Reviewer #1

The authors have provided additional information that addresses the most serious problems of the data presented. By providing current efficiency, light output etc. for each frequency the reader can now get a much better understanding of the performance that has been achieved so far. Quoting efficiency with a brightness is also a marked improvement and for high frequency drive, reasonable efficiency is achieved at useful brightness. Efficiency at mains frequency is lower, but at least provides a starting point.

Whilst I still have great difficulty envisaging real use of three phase driving of such devices, as the most serious concerns have been addressed, I do not object to publication of the paper.

We thank the reviewer for useful suggestions on improving the quality of our manuscript.

We will focus on the further developments using different luminescent materials (such as organic, polymer, perovskite or quantum dots EL materials), modification of the functional layers (such as retroreflective electrodes, dielectric layer modification and electric field enhancing material doped phosphor layer) or advanced fabrication methods (such as photolithography) applying to such TPEL architectures to extend their potential applications and make the technology viable.

Reviewer #4

AUTHORS have accounted the comments of reviewer 1. I am satisfied with added new Figs on luminance and current efficiencies at 1 Khz and 10 Khz.

And with all other additions and changes. Now manuscript can be accepted for publication.

We thank the reviewer for useful suggestions on improving the quality of our manuscript. And we thank the reviewer for this positive comment.